human–computer interaction/psychology

video games, human motivation, well-being

**Author for correspondence:**
Andrew K. Przybylski
e-mail: andy.przybylski@oii.ox.ac.uk

[†]NJ, MV, & AKP declare equal contribution to this work.

# Video game play is positively correlated with well-being

Niklas Johannes[†], Matti Vuorre[†]
and Andrew K. Przybylski[†]

Oxford Internet Institute, University of Oxford, Oxford, UK

NJ, 0000-0001-6612-2842; MV, 0000-0001-5052-066X;
AKP, 0000-0001-5547-2185

People have never played more video games, and many stakeholders are worried that this activity might be bad for players. So far, research has not had adequate data to test whether these worries are justified and if policymakers should act to regulate video game play time. We attempt to provide much-needed evidence with adequate data. Whereas previous research had to rely on self-reported play behaviour, we collaborated with two games companies, Electronic Arts and Nintendo of America, to obtain players' actual play behaviour. We surveyed players of *Plants vs. Zombies: Battle for Neighborville* and *Animal Crossing: New Horizons* for their well-being, motivations and need satisfaction during play, and merged their responses with telemetry data (i.e. logged game play). Contrary to many fears that excessive play time will lead to addiction and poor mental health, we found a small positive relation between game play and affective well-being. Need satisfaction and motivations during play did not interact with play time but were instead independently related to well-being. Our results advance the field in two important ways. First, we show that collaborations with industry partners can be done to high academic standards in an ethical and transparent fashion. Second, we deliver much-needed evidence to policymakers on the link between play and mental health.

## 1. Introduction

Video games are an immensely popular and profitable leisure activity. Last year, the revenues of the games industry were larger than the film industry's [1] and the number of people who report playing games has never been higher [2]. Across the globe, the rise of games as a dominant form of recreation and socializing has raised important questions about the potential effect of play on well-being. These questions concern players, parents, policymakers and scholars alike: billions of people play video games, and if this activity has positive or negative effects

on well-being, playing games might have worldwide health impacts. Therefore, empirically understanding how games might help or harm players is a top priority for all stakeholders. It is possible games are neutral with respect to health and enacting policies that unnecessarily regulate play would restrict human rights to play and freedom of expression [3]. Decisions on regulating video games, or promoting it as a medium for bolstering health, thus come with high stakes and must not be made without robust scientific evidence.

Unfortunately, nearly three decades of research exploring the possible links between video games and negative outcomes including aggression, addiction, well-being and cognitive functioning have brought us nowhere near a consensus or evidence-based policy because reliable, reproducible and ecologically valid studies are few and far between (e.g. [4,5]). In recent years, researchers and policymakers have shifted focus from concerns about violent video games and aggression (e.g. [6]) to concerns about the association between the amount, or nature, of the time people spend playing video games and well-being (e.g. in the UK [7]). In other words, they are interested in the effect of game play behaviours on subjective well-being and by extension mental health. Yet, instead of measuring such behaviour directly, research has relied on self-reported engagement. Historically, this methodological decision has been taken on practical grounds: first, self-report is a relatively easy way to collect data about play. Second, the video games industry has in the past hesitated to work with independent scientists. As time has gone on, it has become increasingly clear that defaulting to self-report is not tenable. Recent evidence suggests self-reports of digital behaviours are notoriously imprecise and biased, which limits the conclusions we can draw from research on time spent on video games and well-being [8,9].

The lack of accurate behavioural data represents a formidable shortcoming that deprives health policymakers of the high-quality evidence they require to make informed decisions on possible regulations to the video games industry [10]. A range of solutions have been proposed including active and passive forms of online engagement [11] and measuring engagement using device telemetry (i.e. logged game play) [12,13]. Therefore, there is a need for directly measured video game behaviour to inform policymakers. To obtain such data, researchers must collaborate, in a transparent and credible way, with industry data scientists who can record objective measures of video game engagement. In this paper, we detail such a collaboration and report our investigation of the relation between the actual time people devote to playing a game and their subjective sense of well-being. We believe our study addresses the primary impediment to past research, delivers high-quality evidence that policymakers require, and provides a template for transparent, robust and credible research on games and health.

## 1.1. Video game behaviour

Globally speaking, the most contentious debates surrounding the potential effects of video game engagement are focused on the mental health of players. For example, the American Psychiatric Association did not identify any psychiatric conditions related to video games in the Diagnostic and Statistical Manual of Mental Disorders (DSM-5), but it does recommend Internet Gaming Disorder as a topic for further research [14]. The World Health Organization adopted a more definitive approach and included Gaming Disorder in the International Classification of Diseases (ICD-11), emphasizing excessive play time as a necessary component [15]. In sharp contrast, the US Food and Drug Administration recently approved the use of a so-called 'serious video game' for treatment of children with attention deficit hyperactivity disorder, providing some evidence that there are mental health benefits of some kinds of play time [16]. These examples illustrate the central role video game engagement plays as a potential public health issue.

Given this, it is critical to understand that the quality of the evidence underlying possible classifications of video game play as potentially psychopathological has been criticized strongly. Many experts have argued that there is insufficient evidence that gaming disorder definitions and diagnostic tools meet clinical standards [15,17–22]. Excessive use has been flagged as a key criterion for many gaming disorder definitions, yet researchers exclusively operationalize excessive use by way of self-reported estimates. This is an important shortcoming, as an increasing number of scholars are now aware that self-reported behaviour is a poor predictor of actual behaviour, particularly for technology use (e.g. [8,23,24]). Self-reported video game play is thus an unsuitable proxy of actual video game play—yet researchers and those advising health bodies are depending on self-reports for diagnosis and policy decisions (e.g. [21]).

Although there have been calls for more direct measures of video game behaviour, these efforts have stalled because scientists do not have the resources or access to data necessary for independent scientific research. For example, on the issue of social media use and well-being, a UK parliamentary select

committee called on 'social media companies to make anonymized high-level data available, for research purposes' in January 2019 [25]. A year later, another committee report on addictive and immersive digital technologies recommended that government 'require games companies to share aggregated player data with researchers' [26]. There is a need for collaborations between games companies and independent scientists, but we are unaware of any successful collaborations investigating player well-being. Game developers have in-house expertise in directly measuring video game engagement via telemetry—the automated logging of users' interaction with content. But so far, efforts have been futile to connect with scientists who have experience in combining such telemetry data with methods that assess subjective well-being (e.g. surveys or experience sampling) and it is not clear if the data, collected for commercial purposes, could be applied to scientific ends.

Collaboration with industry partners not only has the promise to make objective player behaviours accessible for independent analysis; it also provides an opportunity to address a related problem which has plagued games research for decades: a lack of transparency and rigour. Much research in the quantitative social sciences does not share data for others to independently verify and extend findings (e.g. [27]). Sharing resources and data contribute to a more robust knowledge base [28,29]. It also gives other scientists, the public and policymakers the opportunity to better judge the credibility of research [30,31]. A lack of transparency allows selective reporting and thus contributes to unreliable findings that regularly fail to replicate (e.g. [32–34]). Work by Elson & Przybylski [35] showed that this issue arises regularly in research focused on the effects of technology, including in video games research. Carras et al. [19] summarized systematic reviews on gaming disorder and found a high degree of selective reporting in the literature. To increase public trust in their findings, scientists have an obligation to work as transparently as possible, particularly when they collaborate with industry [36]. Greater transparency will provide a valuable tool for informing policy [37] and the heated academic debates that surround the global health impacts of games.

## 1.2. Video game behaviour and well-being

Research and policymakers have been interested in a wide range of mental health outcomes of video game play. Mental health comprises both negative mental health (e.g. depression) and positive mental health. Positive mental health can be further divided into emotional well-being (i.e. the affective component) and evaluative well-being (i.e. the cognitive component) [38]. Nearly all non-experimental studies examining the links between video games and mental health rely on subjective, self-reported estimates of video play time, either by players themselves or by parents. For example, Maras et al. [39] found a sizeable positive correlation between video game play time and depression in a large sample of Canadian adolescents. The focus of research is often on excessive or problematic video game use, routinely reporting positive correlations between problematic video games and mental health problems in both cross-sectional (e.g. [40]) and longitudinal designs (e.g. [41]).

Because self-reported technology use has shown to be a poor proxy of actual behaviour, such associations will necessarily be biased (e.g. [8]). The same caveat holds for research reporting both positive (e.g. [42]) and nonlinear (e.g. [43]) associations between video game play time and psychological functioning. For example, studies suggest that self-reported technology use can lead to both overestimates and underestimates of the association with well-being compared to directly logged technology use [44–46]. Therefore, our scientific understanding of video game effects is limited by our measures. In other words, the true association could be positive or negative, small or large, irrelevant or significant.

A handful of efforts have combined server logs with survey data [47]. However, these studies mainly used a network approach, modelling offline to online dynamics in leadership [48] and friendship formation in games [49]. Studies combining objective play and well-being are lacking. We need accurate, direct measures of play time to resolve the inconsistencies in the literature on well-being and to ensure the study of games and health is not as fruitless as the study of games and aggression [5].

Whereas the perceptions of players in recalling their video game play time can introduce bias, a decade of research indicates perceptions of the psychological affordances provided by games are important to player experiences in games. According to self-determination theory, any activity whose affordances align with the motivations of people will contribute to their well-being [50]. Motivations can be intrinsic, driven by people's interests and values which result in enjoyment, or extrinsic, inspired by rewards or a feeling of being pressured to do an activity. If an activity also satisfies basic psychological needs for competence, relatedness and autonomy, people will find the activity more motivating, enjoyable and immersive—ultimately leading to higher well-being.

The interplay of the affordances of video games, motivation and needs has shown to be important for subjective well-being. If a game satisfies basic needs people will experience more enjoyment and higher well-being [51]. Conversely, if those needs are not met, frustrated, or play is externally motivated, it is associated with lower psychological functioning [52]. In other words, how play time relates to well-being probably depends on players' motivations and how the game satisfies basic needs. Player experience would thus moderate the association between play time and well-being: if players are intrinsically motivated and experience enjoyment during play, play time will most likely be positively associated with well-being [53,54]. By contrast, when players only feel extrinsic motivation and feel pressured to play, play time might have negative effects on well-being. Such a mechanism aligns well with a recent review that concludes that motivations behind play are likely to be a crucial moderator of the potential effect of play time on well-being [55,56]. However, it is unclear whether such a mechanism only holds true for self-reported play time and perceptions, or whether self-reported perceptions interact with directly measured play time.

## 1.3. This study

In this study, we investigate the relations between video games and positive mental health, namely affective well-being of players (from here on called well-being). We collaborated with two industry partners, Electronic Arts and Nintendo of America, and applied an approach grounded in an understanding that subjective estimates of play time are inaccurate and the motivational experiences of player engagement are important to well-being. To this end, we surveyed players of two popular video games: *Plants vs. Zombies: Battle for Neighborville* and *Animal Crossing: New Horizons*. Our partners provided us with telemetry data of those players. The data allowed us to explore the association between objective play time and well-being, delivering a much-needed exploration of the relation between directly measured play behaviour and positive mental health. We also explored the role of player motivations in this relation, namely whether feelings of autonomy, relatedness, competence, enjoyment and extrinsic motivation interacted with play time.

In the light of calls for more transparency in the Social Sciences (e.g. [31]), we aimed for a transparent workflow to enable others to critically examine and build upon our work. We, therefore, provide access to all materials, data and code on the Open Science Framework (OSF) page of this project (https://osf.io/cjd6z/). The analyses are documented at https://digital-wellbeing.github.io/gametime/. This documentation and the code have been archived on the OSF at https://doi.org/10.17605/OSF.IO/5EF8H.

# 2. Method

## 2.1. Participants and procedure

For this project, we combined objective game telemetry data with survey responses. We did not conduct *a priori* power analyses. Instead, we followed recent recommendations and aimed to collect as many responses as we had resources for [57]. We surveyed the player base of two popular games: *Plants vs. Zombies: Battle for Neighborville* (PvZ) and *Animal Crossing: New Horizons* (AC:NH).

We designed a survey measuring players' well-being, self-reported play and motivations for play, and discussed the survey structure with Electronic Arts. Electronic Arts (EA) programmed and hosted the survey on *Decipher*, an online survey platform, and sent invite emails to adult (at least 18 years old) PvZ players in the US, Canada and the UK. The survey was translated to French for French-speaking Canadians. Participants received an invitation to participate in the survey on the email address they had associated with their EA account. The email invited them to participate in a research project titled 'Understanding Shifting Patterns of Videogame Play and Health Outcomes'. Participants were informed that the aim of the study was to investigate how people play video games and how they feel over time. We also informed them that Electronic Arts would link their survey responses to their play data. Further, the study information explained that the research team would act independently of Electronic Arts in data analysis and scientific reporting. We obtained ethical approval from our institute (SSH_OII_CIA_20_043) and all respondents gave informed consent. Participants could halt their participation at any time and did not receive compensation for their participation.

Electronic Arts then pulled telemetry game data of players that got invited in the first wave of data collection. They matched telemetry data with survey invitations by a securely hashed player ID. Afterwards, they transferred both the survey and the telemetry datasets to the researchers. Neither

dataset contained personally identifiable information, only a hashed player ID that we used to link survey and telemetry data. Electronic Arts sent out the invitations in two waves. The first wave happened in early August 2020 and was sent to 50 000 player (response window: 48 h).[1] We inspected the data quality of their telemetry and survey responses and checked whether the data were suitable to address our research questions. After confirming that the data were suitable and that we could join the telemetry and survey information, Electronic Arts sent out a second wave of invitations to 200 000 PvZ players from the same population at the end of September 2020 (response window: 96 h). In total, 518 PvZ players (approx. 0.21% response rate) finished the survey ($M_{age}$ = 35, s.d.$_{age}$ = 12; 404 men, 94 women, two other, 17 preferred not to disclose their gender), of whom 471 had matching telemetry data.

For *Animal Crossing: New Horizon* (AC:NH) players, the procedure was similar. We hosted the survey with *formr* [58], an open source survey tool, and Nintendo of America sent invitations with survey links to a 342 825 adult players in the US on 27 October. The survey was identical to the one sent to PvZ players except for aesthetic differences. The response window was 7 days and in total, 6011 players responded (1.75% response rate; $M_{age}$ = 31, s.d.$_{age}$ = 10; 3124 men, 2462 women, 153 other, 88 preferred not to disclose their gender). We then provided the hashed IDs of the survey respondents to Nintendo of America, who sent us the telemetry data for those players, of whom 2756 had telemetry within the two-week window. Neither the survey nor the telemetry data had any personally identifiable information. We followed the same workflow as described above and linked survey responses with play data.

## 2.2. Measures

### 2.2.1. Well-being

We assessed well-being with the validated scale of positive and negative experiences (SPANE, [59]), which measures the affective dimension of well-being [38]. We asked respondents to think about how they had been feeling in the past two weeks and report how often they experienced each of six positive and six negative feelings. Respondents could indicate the frequency of experiencing those feelings on a scale from 1 (*Very rarely or never*) to 7 (*Very often or always*). We then took the mean of the positive feelings and negative feelings and subtracted the negative affect mean score from the positive affect mean score to obtain a measure of well-being (figure 1).

### 2.2.2. Player experience and need satisfaction

We assessed player experiences and motivations with the player experience and need satisfaction scale (PENS, [51]), which has recently been validated [60]. We asked respondents to rate items reflecting on when they had been playing PvZ/AC:NH in the past two weeks on a scale from 1 (*Strongly disagree*) to 7 (*Strongly agree*). The scale consisted of five subscales. Participants reported their sense of autonomy on three items such as 'I experienced a lot of freedom in [PvZ/AC:NH]'; their sense of competence on three items such as 'I felt competent at [PvZ/AC:NH]'; their sense of relatedness on items such as 'I found the relationships I formed in [PvZ/AC:NH] fulfilling', but only when they reported to have played with others, either online or in couch co-op, in the past two weeks. They also reported their enjoyment with four items such as 'I think [PvZ/AC:NH] was fun to play' and their extrinsic motivation on four items such as 'I played [PvZ/AC:NH] to escape'.

### 2.2.3. Self-reported play

Participants also reported how much total time they estimated to have spent playing the game in the past two weeks on two open numerical fields, where they could report hours and minutes (figure 2). We transformed both to total time in hours.[2] For PvZ, the *Decipher* survey platform restricted the maximum time players could report to 40 h 59 min. Such large values were rare and affected only a handful of participants. The AC:NH time scale was unrestricted.

---

[1]Due to an error, 15 individuals in total participated after the response window in waves 1 and 2.

[2]We also measured other variables that we do not report on here. For example, respondents reported their estimates of how their play changed compared to others in the past two weeks. The full list of measures is in the online materials. The data for these measures are on the OSF project of this manuscript.

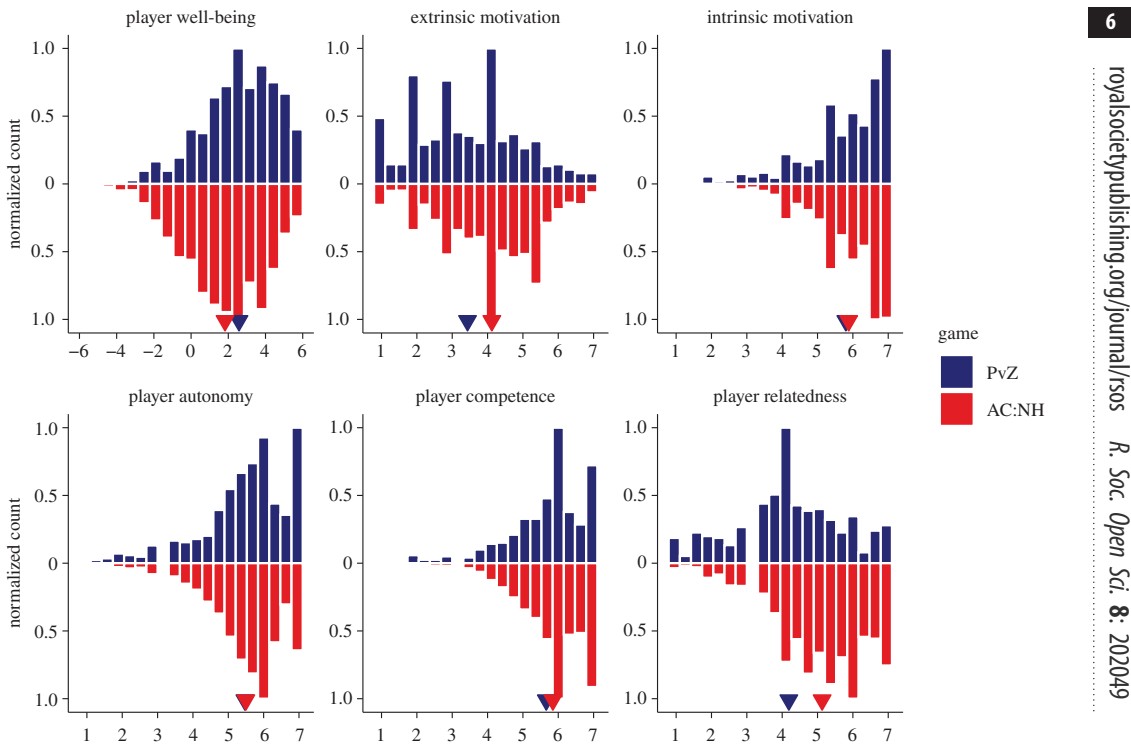

**Figure 1.** Histograms of central variables. The *y*-axis indicates counts of responses in each bin, scaled to the bin with greatest number of responses. Top frequencies indicate PvZ players' responses, bottom frequencies indicate AC:NH players' responses. Small triangles indicate means.

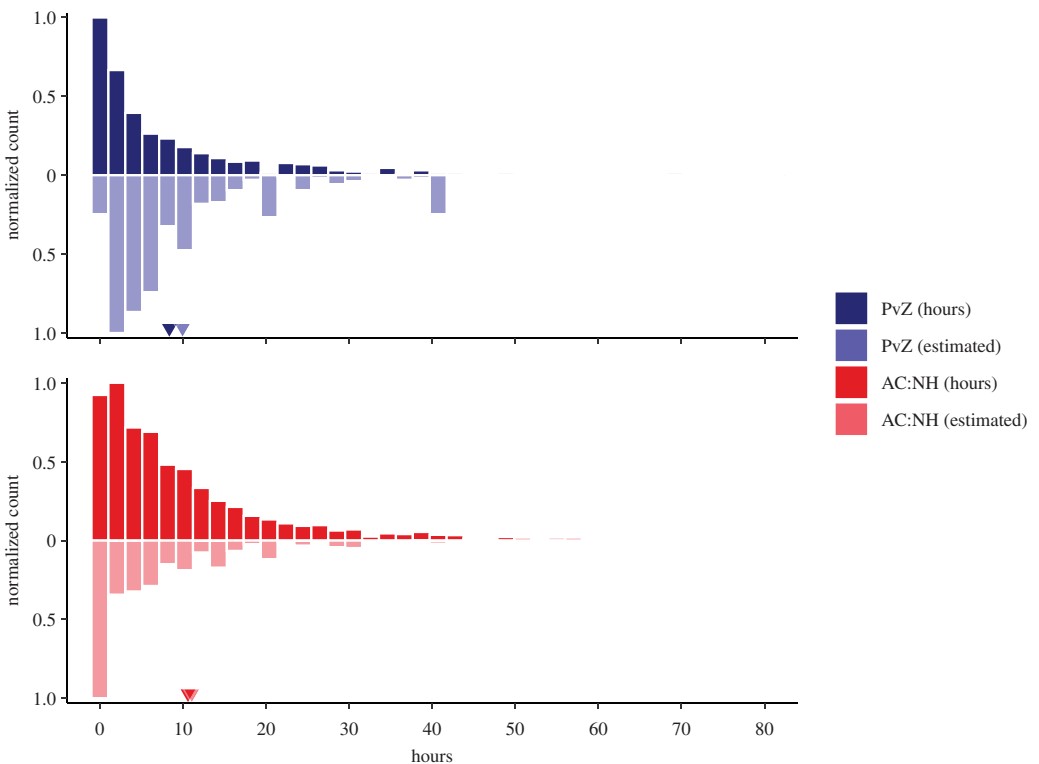

**Figure 2.** Histograms of actual play time (solid; top) and subjective estimates of play time (light; bottom) for both games. Small triangles indicate means over participants who had data for both variables. The *x*-axis in this figure is truncated at 80 h to make the bulk of the values easier to discern. 65 AC:NH time values (7 [0.3%; max = 99.8] actual, 58 [1.0%; max = 161.0] estimated) were above this cut-off and are therefore not shown on this figure.

### 2.2.4. Telemetry

Telemetry data for both games were available in different levels of granularity. For both PvZ and AC:NH, the games companies provided game sessions per player over the two-week window before finishing the survey. Each game session had a start time and an end time. For example, when a player turned on their console, launched PvZ and entered the game hub world (where they can select what type of level or mode to play), opening the hub world counted as the start time. However, determining an end time can be difficult. Players could immediately play another round, return to the hub world, take a break while leaving the game on etc. Therefore, there were instances where game sessions for a given player overlapped (e.g. two game sessions had the same start time, but different end times). In such cases, we condensed multiple overlapping game sessions into one game session, taking the game start time that the sessions shared and the last end time. As a result, players had multiple unique game sessions without overlap. Afterwards, we aggregated the durations of all game sessions per player to obtain the total objective time they spent playing the game in the two-week window before they filled out the survey. In the case of AC:NH, Nintendo of America provided telemetry including game session start and end times, as well as session durations. Start and end times were not always accurate for the same reasons as with PvZ, but we used them in order to only use the session information from the two weeks preceding the survey. However, the session durations were verified by the Nintendo of America Team. Therefore, we aggregated durations in the two weeks preceding the survey per participant for AC:NH (figure 2). Readers can find more details on data processing on https:// digital-wellbeing.github.io/gametime/.

PvZ telemetry featured fine-grained records of game events. The data contained information on the game mode of a game session (e.g. online or split screen), the levels a player played in during a game session, and the game type (i.e. single player versus multiplayer). Furthermore, there were indicators of how a player fared during a game session, such as total kill counts, death counts, scores, total damage dealt, shots fired and hit, and critical hit counts. PvZ telemetry also contained information on a player's progression, such as when a player gained a level, how much XP they gained, and when they gained a prestige level. Last, there were measures of social interactions with other players, such as what in-game gestures a player used and when they became in-game friends with other players. We did not analyse these data or report on them here. Readers can find them on the OSF page of this article.

## 2.3. Statistical analysis

Before analysis, we first excluded data from players who gave the same response to all SPANE and motivations items (PvZ: 1 [0.2%]; AC:NH: 8 [0.1%]). This so-called straight lining is an indicator of poor data quality [61]. We then identified as outliers all observations that were more than six standard deviations away from the variable's mean. We aimed to exclude as few data points as possible, which is why we did not follow the common rule of thumb of three standard deviations and only identify truly extreme, implausible values. We replaced those extreme values with missing values to not bias resulting analyses. For PvZ, 1 (0.2%) objective play time values were excluded. For AC: NH, 12 (0.2%) actual and 40 (0.7%) estimated play time values were excluded. In the models reported below, play time refers to units of 10 h, and the well-being and motivation variables were standardized. Note that the sample sizes per analysis will differ because of missing values either due to exclusions, not having telemetry data, or not having played socially (i.e. no responses on relatedness). We conducted all statistical analyses with R (v. 4.0.3, [62]).

# 3. Results

## 3.1. Play time and affective well-being

We first focused on the relationships among objective and subjective play time, and well-being. The actual and estimated play times are shown in figure 2. On average, participants overestimated their play time (PvZ: $M = 1.6$, s.d. $= 11.8$; AC:NH: $M = 0.5$, s.d. $= 15.8$).

Then, to better understand how actual and estimated play times were related, we regressed the subjective estimates of play time on objective play time. This correlation was positive (PvZ: $\beta = 0.34$, 95%CI [0.27, 0.41], $R^2 = 0.15$, $N = 469$; AC:NH: $\beta = 0.49$, 95%CI [0.45, 0.54] $R^2 = 0.16$, $N = 2714$;

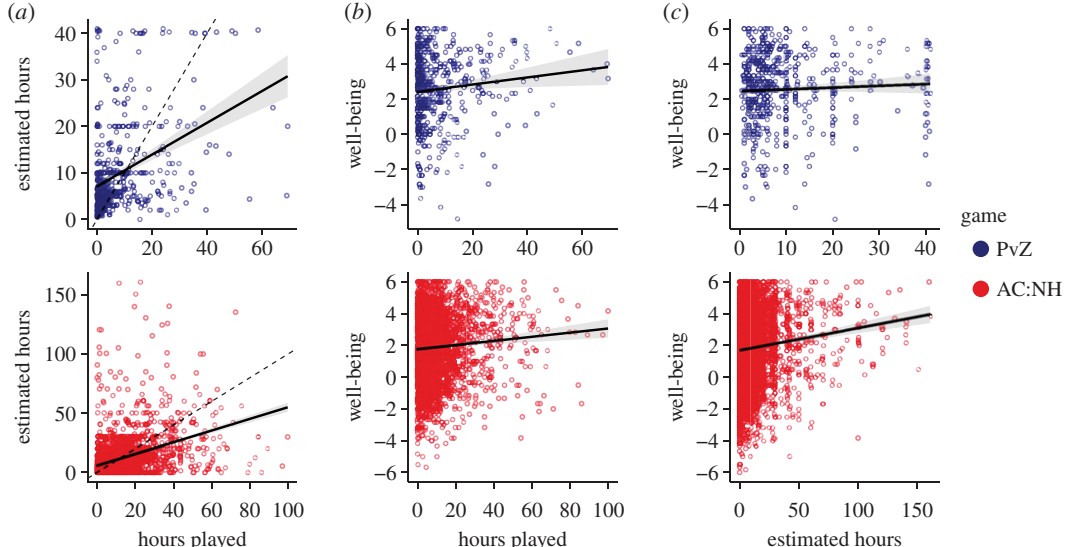

**Figure 3.** (*a*) Relationship between the objective time played and participants' estimates of the time spent playing. Points indicate individuals, solid line and shade are the regression line and its 95%CI. The dashed line indicates a perfect relationship. (*b*) Relationship between the objective time spent playing, and well-being. (*c*) Relationship between participants' estimated time spent playing and well-being.

figure 3*a*). Readers can find the full regression tables on https://digital-wellbeing.github.io/gametime/analyses.html#analyses.

Next, we investigated the relationship between play time and well-being, as measured by SPANE. The relationship for objective play time was positive and significant (PVZ: $\beta = 0.10$, 95%CI [0.02, 0.18], $R^2 = 0.01$, $N = 468$; AC:NH: $\beta = 0.06$, 95%CI [0.03, 0.09], $R^2 = 0.01$, $N = 2537$; figure 3*b*). Thus, with each additional 10 h of playing the game, players reported a 0.02–0.18 (PvZ; AC:NH: 0.03–0.09) standard deviation increase in well-being. The relation to subjective time estimates was only significant in AC: NH, but not PVZ (PVZ: $\beta = 0.05$, 95%CI [−0.04, 0.14], $R^2 = 0.00$, $N = 516$; AC:NH: $\beta = 0.07$, 95%CI = [0.05, 0.08], $R^2 = 0.01$, $N = 5487$; figure 3*c*).

We also investigated possible nonlinear relations between play time and well-being, because the relationship might be different between people who play a great deal and people who only play a little. To do so, we compared a generalized additive model [63] of well-being with and without a smooth term for actual play time, separately for each game and subjective/objective time. None of the AIC differences were greater than 1, indicating that the linear models were adequate descriptions of the associations between play time and well-being.[3] Together, these results showed that subjective estimates of play time are, at best, uncertain indicators of true engagement, and that the latter has a positive but small relation to well-being.

## 3.2. Well-being and motivation

We then turned our attention to the potentially moderating roles that needs and motivations might have in the relation between play time and well-being. That is, the relation between play time and well-being might vary according to how players experienced play: if players experienced intrinsic motivations and need satisfaction during play, we would expect a more positive relationship between play time and well-being compared to players who experienced less intrinsic motivation and need satisfaction during play. By contrast, if players felt extrinsically motivated during play (i.e. pressured to play), their play time might have a negative relation to well-being, compared to players who experienced less of an extrinsic motivation during play.

To study this idea, we specified a multiple linear regression model whereby well-being (standardized) was predicted from the five PENS subscales (sense of autonomy, competence, relatedness and extrinsic and intrinsic [enjoyment] motivations; standardized) and play time (in units

---

[3]There was some evidence from this model comparison suggesting that the association between subjective play time estimates and well-being is better described by a curvilinear function than a linear fit, but the differences between models were small.

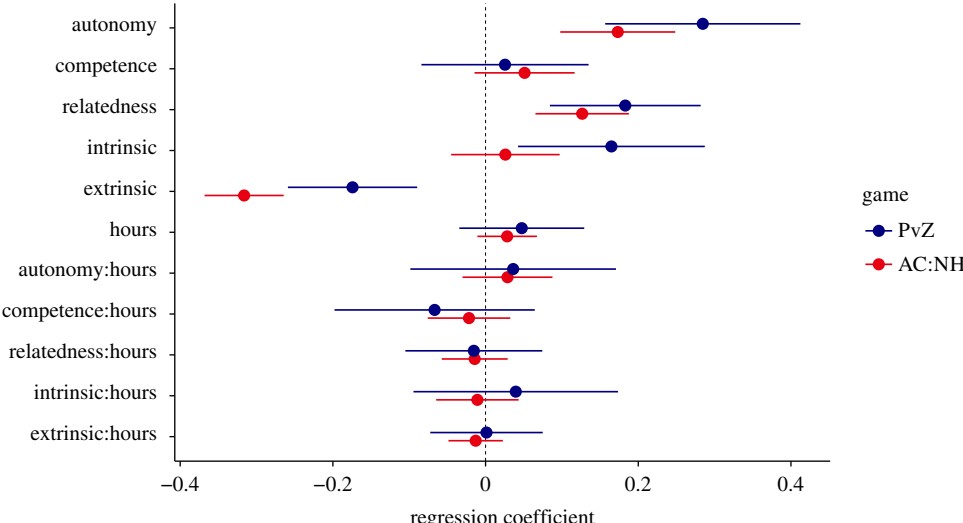

**Figure 4.** Parameter estimates from multiple linear regression model predicting affective well-being. Parameters with a colon indicate interaction effects (e.g. autonomy:hours indicates the degree to which experiences of autonomy moderate the relation between play time and affective well-being). Error bars indicate 95%CIs.

of 10 h). Critically, we also included the two-way interactions between play time and motivation variables to study how the relations between play time and well-being might be moderated by the experienced needs and motivations. We standardized all PENS scale scores. Results from this model are shown in figure 4 (PvZ: $R^2 = 0.29$, $N = 404$; AC:NH: $R^2 = 0.15$, $N = 1430$). First, conditional on objective play time, experiences of autonomy, competence, relatedness and intrinsic motivation (enjoyment) positively predicted affective well-being, whereas extrinsic motivations were negatively associated with it. However, only autonomy, relatedness and extrinsic motivations were significant predictors in both games. In this model, conditional on the experienced needs and motivations (which were standardized), time spent playing still positively predicted subjective well-being, but the relation was not significant.

Next, we investigated the interaction effects, which showed how the relation between play time and well-being varied with different levels of experienced need satisfaction and motivation. The results indicated no consistent pattern in the moderating roles of motivational experiences on how play time related to well-being. For example, the more experiences of autonomy a player experienced, the more positively was their play time related to well-being, but this interaction was not significant in either game. Even with the larger AC:NH sample, no significant interactions were detected. However, experienced autonomy and relatedness emerged as consistent predictors of well-being, and extrinsic motivations as a negative predictor. Taken together, these results suggested that players' in-game motivational experiences can contribute to affective well-being, but they do not affect the degree to which play time relates to well-being.

## 4. Discussion

How is video game play related to the mental health of players? This question is at the heart of the debate on how policymakers will act to promote or to restrict games' place in our lives [7]. Research investigating that question has almost exclusively relied on self-reports of play behaviour, which are known to be inaccurate (e.g. [8]). Consequently, we lack evidence on the relation between play time and mental health that is needed to inform policy decisions. To obtain reliable and accurate play data, researchers must collaborate with industry partners. Here, we aimed to address these shortcomings in measurement and report a collaboration with two games companies, Electronic Arts and Nintendo of America, combining objective measures of game behaviour (i.e. telemetry) with self-reports (i.e. survey) for two games: *Plants vs. Zombies: Battle for Neighborville* and *Animal Crossing: New Horizons*. We also explored whether the relation between play time and well-being varies with players' need satisfaction and motivations. We found a small positive relation between play time and well-being for both games. We did not find evidence that this relation was moderated by need satisfactions and

motivations, but that need satisfaction and motivations were related to well-being in their own right. Overall, our findings suggest that regulating video games, on the basis of time, might not bring the benefits many might expect, though the correlational nature of the data limits that conclusion.

Our goal was to investigate the relation between play time, as a measure of actual play behaviour, and subjective well-being. We found that relying on objective measures is necessary to assess play time: although there was overlap between the amount of time participants estimated to have played and their actual play time as logged by the game companies, that relation was far from perfect. On average, players overestimated their play time by 0.5 to 1.6 hours. The size of that relation and the general trend to overestimate such technology use are in line with the literature, which shows similar trends for internet use [24] and smartphone use [8,23]. Therefore, when researchers rely on self-reports of play behaviour to test relations with mental health, measurement error and potential bias will necessarily lead to inaccurate estimates of true relationships. Previous work has shown that using self-reports instead of objective measures of technology use can both inflate [45,46] or deflate effects [44]. In our study, associations between objective play time and well-being were larger than those between self-reported play time and well-being. Had we relied on self-reports only, we could have missed a potentially meaningful association.

Players who objectively played more in the past two weeks also reported to experience higher well-being. This association aligns well with literature that emphasizes the benefits of video games as a leisure activity that contributes to people's mental health [42]. Because our study was cross-sectional, there might also be a self-selection effect: People who feel good might be more inclined to pick up their controller. Such a view aligns well with research that shows reciprocal relations between media use and well-being [64,65]. Equally plausible, there might be factors that affect both game play time and well-being [66,67]. For example, people with high incomes are likely to be healthier and more likely to be able to afford a console/PC and the game.

Even if we were to assume that play time directly predicts well-being, it remains an open question whether that effect is large enough to matter for people's subjective experience. From a clinical perspective, it is probably the effect is too small to be relevant for clinical treatments. Our effect size estimates were below the smallest effect size of interest for media effects research that Ferguson [68] proposes. For health outcomes, Norman and colleagues [69] argue that we need to observe a large effect size of around half a standard deviation for participants to feel an improvement. In the AC:NH model, 10 h of game play were associated with a 0.06 standard deviation increase in well-being. Therefore, a half standard deviation change would require approximately 80 h of play over the two weeks (translating to about 6 h per day). However, Anvari and Lakens demonstrated that people might subjectively perceive differences of about a third of a standard deviation on a measure of well-being similar to ours [70], suggesting that approximately three and a half hours of play might be associated with subjectively felt changes in well-being. Nevertheless, it is unclear whether typical increases in play go hand in hand with perceivable changes in well-being. However, even small relations might accumulate to larger effects over time, and finding boundary conditions, such as time frames under which effects are meaningful, is a necessary next step for research [71]. Moreover, we only studied one facet of positive mental health, namely affective well-being. Future research will need to consider other facets, such as negative mental health.

Although our data do not allow causal claims, they do speak to the broader conversation surrounding the idea of video game addiction (e.g. [15]). The discussion about video games has focused on fears about a large part of players becoming addicted [14,21]. Given their widespread popularity, many policymakers are concerned about negative effects of play time on well-being [7]. Our results challenge that view. The relation between play time and well-being was positive in two large samples. Therefore, our study speaks against an immediate need to regulate video games as a preventive measure to limit video game addiction. If anything, our results suggest that play can be an activity that relates positively to people's mental health—and regulating games could withhold those benefits from players.

We also explored the role of people's perceptions in the relation between play time and well-being. Previous work has shown that gamers' experience probably influences how playing affects mental health [51,52]. We explored such a possible moderation through the lens of self-determination theory [50]: We investigated whether changes in need satisfaction, enjoyment and motivation during play changed the association between play time and well-being. We found no evidence for moderation. Neither need satisfaction, nor enjoyment, nor extrinsic motivation significantly interacted with play time in predicting well-being. However, conditional on play time, satisfaction of the autonomy and relatedness need, as well as enjoyment were positively associated with well-being. Extrinsic

motivation, by contrast, was negatively associated with well-being. These associations line up with research demonstrating that experiencing need satisfaction and enjoyment during play can be a contributing factor to user well-being, whereas an extrinsic motivation for playing probably does the opposite (e.g. [56]).

Although we cannot rule out that these player experiences had a moderating role, the estimates of the effect size suggest that any moderation is likely to be too small to be practically meaningful. In other words, our results do not suggest that player experience modulates the relation between play time and well-being, but rather contributes to it independently. For example, players who experience a high degree of relatedness during play will probably experience higher well-being, but a high degree of relatedness is unlikely to strengthen the relation between play time and well-being. Future research, focused on granular in-game behaviours such as competition, collaboration and advancement will be able to speak more meaningfully to the psychological affordances of these virtual contexts.

Conditional on those needs and motivations, play time was not significantly related to well-being anymore. We are cautious not to put too much stock in this pattern. A predictor becoming not significant when controlling for other predictors can have many reasons. Need satisfaction and motivations might mediate the relation between play time and well-being; conditioning on the mediator could mask the effect of the predictor [67]. Alternatively, if play time and player experiences are themselves related, including them all as predictors would result in some relations being overshadowed by others. We need empirical theory-driven research grounded in clear causal models and longitudinal data to dissect these patterns.

## 4.1. Limitations

We are mindful to emphasize that we cannot claim that play time causally affects well-being. The goal of this study was to explore whether and how objective game behaviour relates to mental health. We were successful in capturing a snapshot of that relation and gaining initial insight into the relations between video games and mental health. But policymakers and public stakeholders require evidence which can speak to the trajectory of play and its effect over time on well-being. Video games are not a static medium; both how we play and discuss them is in constant flux [72]. To build on the work we present here, there is an urgent need for collaborations with games companies to obtain longitudinal data that allow investigating all the facets of human play and its effects on well-being over time.

Longitudinal work would also address the question of how generalizable our findings are. We collected data during a pandemic. It is possible the positive association between play time and well-being we observed only holds during a time when people are naturally playing more and have less opportunity to follow other hobbies. Selecting two titles out of a wide range of games puts further limitations on how generalizable our results are. Especially *Animal Crossing: New Horizons* is considered a casual game with little competition. Therefore, although those two titles were drawn from different genres, we cannot generalize to players across all types of games [73]. The results might be different for more competitive games. Different games have different affordances [74] and, therefore, likely different associations with well-being. To be able to make recommendations to policymakers on making decisions across the diverse range of video games, we urge video game companies to share game play data from more titles from different genres and of different audiences. Making such large-scale data available would enable researchers to match game play with existing cohort studies. Linking these two data sources would enable generalizable, causal tests of the effect of video games on mental health.

Another limiting factor on the confidence in our results is the low response rate observed in both of our surveys. It is possible that various selection effects might have led to unrepresentative estimates of well-being, game play, or their relationship. Increasing response rates, while at the same time ensuring samples' representativeness, remains a challenge for future studies in this field.

Our results are also on a broad level—possibly explaining the small effect sizes we observed. When exploring effects of technology use on well-being, researchers can operate on several levels. As Meier & Reinecke [75] explain, we can choose to test effects on the device level (e.g. time playing on a console, regardless of game), the application level (e.g. time playing a specific game), or the feature level (e.g. using gestures in a multiplayer game). Here, we operated on the application level, which subsumes all possible effects on the feature level. In other words, when measuring time with a game, some features of the game will have positive effects; others will have negative effects. Measuring on the application level will thus only give us a view of 'net' video game effects. Assessing game behaviour on a more granular level will be necessary to gain more comprehensive insights and make specific recommendations to policymakers. For that to happen, games companies will need to have

transparent, accessible APIs and access points for researchers to investigate in-game behaviour and its effects on people's mental health. Such in-game behaviours also carry much promise for studying the therapeutic effects of games, for example, as markers of symptom strength in disorders [76]. In rare cases, researchers were able to make use of such APIs [47,49], but the majority of games data are still not accessible. For PvZ, EA provided a variety of in-game behaviours that we did not analyse here. We invite readers to explore those data on the OSF project of this manuscript.

We relied on objective measures of video game behaviour. These measures are superior to self-reported behaviour because they directly capture the variable of interest. However, capturing game sessions on the side of the video game companies comes with its own measurement error. Video game companies cannot perfectly measure each game session. For example, in our data processing, some game sessions had duplicate start and end times (for PvZ) or inaccurate start and end times, but accurate session durations (for AC:NH). Measurement error in logging technology use is a common issue (e.g. [12,77]), and researchers collaborating with industry partners need to understand how these partners collect telemetry. The field needs to embrace these challenges in measurement rather than defaulting to self-reports.

Last, this study was exploratory and we made decisions about data processing and analysis without specifying them *a priori* [78]. Such researcher degrees of freedom can yield different results, especially in the field of technology use and well-being [65,79]. In our process, we were as transparent as possible to enable others to examine and build upon our work [31]. To move beyond this initial exploration of objective game behaviour and well-being to a more confirmatory approach, researchers should follow current best practices: they should preregister their research before collecting data in collaboration with industry partners [80,81], before accessing secondary data sources [82], and consider the registered report format [83,84]. Following these steps will result in a more reliable knowledge base for policymakers.

# 5. Conclusion

Policymakers urgently require reliable, robust and credible evidence that illuminates the influence video games may have on global mental health. However, the most important source of data, the objective behaviours of players, are not used in scientific research. Instead, researchers have needed to fall back on asking players to report their behaviour, a flawed approach which cannot effectively guide policy. In this study, we show that collaborations with industry partners to obtain adequate data are possible. Research with these data can be done to academic standards—ethically and transparently. We are optimistic that collaborations of this sort will deliver the evidence required to advance our understanding of human play and provide policymakers the insights into how they might shape, for good or ill, our health.

Ethics. The study underwent ethical review and received approval from the Central University Ethics Committee of the University of Oxford (SSH_OII_CIA_20_043).

Data accessibility. Readers can find all materials, data and code on the Open Science Framework (OSF) page of this project (https://osf.io/cjd6z/). The analyses are documented at https://digital-wellbeing.github.io/gametime/. This documentation and the code have been archived on the OSF at https://doi.org/10.17605/OSF.IO/5EF8H.

Authors' contributions. Conceptualization: N.J., M.V. and A.K.P. Data Curation: N.J. and M.V. Formal Analysis: N.J. and M.V. Funding Acquisition: A.K.P. Investigation: N.J., M.V. and A.K.P. Methodology: N.J., M.V. and A.K.P. Resources: A.K.P. Software: N.J. and M.V. Supervision: A.K.P. Validation: N.J. and M.V. Visualization: N.J. and M.V. Writing—original draft preparation: N.J., M.V. and A.K.P. Writing—review and editing: N.J., M.V. and A.K.P.

Competing interests. The authors declare no competing interests. The funders had no role in study design, data collection and analysis, decision to publish, or preparation of the manuscript. The industry partners reviewed study design and assisted with data collection, but had no role in data analysis, decision to publish or preparation of the manuscript.

Funding. The research was supported by grants from the Huo Family Foundation and the Economic and Social Research Council (ES/T008709/1) and by in-kind technical contributions by Electronic Arts and Nintendo of America.

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
