## [Peer Review File · Royal Society Open Science]

Review History

RSOS-202049.R0 (Original submission)

Review form: Reviewer 1 (Linda Kaye)

Is the manuscript scientifically sound in its present form?

Yes

Are the interpretations and conclusions justified by the results?

Yes

Is the language acceptable?

Yes

Do you have any ethical concerns with this paper?

No

Have you any concerns about statistical analyses in this paper?

No

Recommendation?

Accept with minor revision (please list in comments)

Comments to the Author(s)

Thank you for providing me the opportunity to review this work. I really like the approach of using objective play time data to correspond to well-being measures. This is a much needed development and contribution in this field. The work has been conducted in a rigorous way and presented well. Thank you to the authors for providing their resources on OSF and the offer of using the supplementary data. I have some minor comments which may support some of the conceptual framing/presentation of the manuscript and overall am highly supportive of this work being published.

#1. The introduction could do with a little more theoretical detail about basic needs theory/SDT as currently player needs is discussed only in passing. What are these and why are they relevant to gaming?

#2. The authors talk about "well-being" and then mention example studies, one including depression. As well-being is a wide construct, some outline of this would be useful ie how are the authors conceptualising it when discussing it within the introductory sections? Also, in the "This study" section, the authors interchange the terminology. In some cases they outline they will be exploring "subjective well-being" and other times "subjective mental health". Are these the same thing? The method suggests that affective wellbeing was measured so some conceptualisation earlier in the paper would be helpful on this.

#3. The analysis discusses that the authors paid attention to the moderating role of needs and motivations between play time and wellbeing. There has been no conceptual framing of this previously and so the intro would benefit from some detail about this. Eg in what way are specific needs likely to be a moderator between these variables? There is a brief description of expecting findings on p11, but no theoretical/empirical basis for this in the intro in the current manuscript.

#4. The current study focuses exclusively on two games as is appropriate given the methodology of garnering specific game server data. The intro prior to this has discussed gaming in very generic terms and I wonder how much of the evidence in the literature review is relevant to the types of games being used in the current work. My understanding from this literature is a lot of what has been discussed in the lit review relates more to studies which have looked at MMOs or assumed "hardcore" forms of gaming (or been very general). The two games in the current study are a simulation game and TPS, both of which seem more tailored to the casual market to some extent. How applicable are the previous findings about gaming and well-being to the specific types of games of the current study? It would be useful for the authors to outline some distinctions here in their rationale-building perhaps to identify that these specific types of games may have been less exclusively focused on in the previous literature (and this itself may be a contribution). This may have implications however, as the links between gaming and wellbeing found here may not preclude the previous work on different types of games and well-being being invalid. The authors do make reference to the issue of game genre in the discussion limitations but it would be useful that this is framed earlier in the paper

Minor

#5. P9-10- would use the term "play time" consistently rather than switch between "game time" and "play time" (see also discussion for this too)

Review form: Reviewer 2 (Michelle Colder Carras)

Is the manuscript scientifically sound in its present form?

Yes

Are the interpretations and conclusions justified by the results?

Yes

Is the language acceptable?

Yes

Do you have any ethical concerns with this paper?

No

Have you any concerns about statistical analyses in this paper?

No

Recommendation?

Accept with minor revision (please list in comments)

Comments to the Author(s)

This landmark paper examines well-being in a large sample of gamers by combining a relatively new source of data, actual behavioral footprint from game developers, with a traditional survey. The background information and justification is very well developed and the methods are strong. Such a difficult analysis is hard to describe with enough clarity that people who aren't familiar with the analysis can understand it, and some suggestions are made here for minor changes that could help that. I also have a slight concern about period effects and the description of the sample that are not listed in the Limitations section. In my opinion, the paper needs only minor revisions before it can be published and set a new standard in the field of video game research. Please see the attached file (Appendix A) for many additional suggestions.

Decision letter (RSOS-202049.R0)

Dear Dr Johannes

The Editors assigned to your paper RSOS-202049 "Video game play is positively correlated with well-being" have now received comments from reviewers and would like you to revise the paper in accordance with the reviewer comments and any comments from the Editors. Please note this decision does not guarantee eventual acceptance.

Please submit your revised manuscript and required files (see below) no later than 21 days from today's (ie 11-Dec-2020) date. Note: the ScholarOne system will 'lock' if submission of the revision is attempted 21 or more days after the deadline. If you do not think you will be able to meet this deadline please contact the editorial office immediately.

on behalf of Dr Joydeep Bhattacharya (Associate Editor) and Essi Viding (Subject Editor)
openscience@royalsociety.org

Reviewer comments to Author:
Reviewer: 1

Comments to the Author(s)

Thank you for providing me the opportunity to review this work. I really like the approach of using objective play time data to correspond to well-being measures. This is a much needed development and contribution in this field. The work has been conducted in a rigorous way and presented well. Thank you to the authors for providing their resources on OSF and the offer of using the supplementary data. I have some minor comments which may support some of the conceptual framing/presentation of the manuscript and overall am highly supportive of this work being published.

#1. The introduction could do with a little more theoretical detail about basic needs theory/SDT as currently player needs is discussed only in passing. What are these and why are they relevant to gaming?

#2. The authors talk about "well-being" and then mention example studies, one including depression. As well-being is a wide construct, some outline of this would be useful ie how are the authors conceptualising it when discussing it within the introductory sections? Also, in the "This study" section, the authors interchange the terminology. In some cases they outline they will be exploring "subjective well-being" and other times "subjective mental health". Are these the same thing? The method suggests that affective wellbeing was measured so some conceptualisation earlier in the paper would be helpful on this.

#3. The analysis discusses that the authors paid attention to the moderating role of needs and motivations between play time and wellbeing. There has been no conceptual framing of this previously and so the intro would benefit from some detail about this. Eg in what way are specific needs likely to be a moderator between these variables? There is a brief description of expecting findings on p11, but no theoretical/empirical basis for this in the intro in the current manuscript.

#4. The current study focuses exclusively on two games as is appropriate given the methodology of garnering specific game server data. The intro prior to this has discussed gaming in very generic terms and I wonder how much of the evidence in the literature review is relevant to the types of games being used in the current work. My understanding from this literature is a lot of what has been discussed in the lit review relates more to studies which have looked at MMOs or assumed "hardcore" forms of gaming (or been very general). The two games in the current study are a simulation game and TPS, both of which seem more tailored to the casual market to some extent. How applicable are the previous findings about gaming and well-being to the specific types of games of the current study? It would be useful for the authors to outline some distinctions here in their rationale-building perhaps to identify that these specific types of games may have been less exclusively focused on in the previous literature (and this itself may be a contribution). This may have implications however, as the links between gaming and wellbeing found here may not preclude the previous work on different types of games and well-being being invalid. The authors do make reference to the issue of game genre in the discussion limitations but it would be useful that this is framed earlier in the paper

Minor

#5. P9-10- would use the term "play time" consistently rather than switch between "game time" and "play time" (see also discussion for this too)

Reviewer: 2

Comments to the Author(s)

This landmark paper examines well-being in a large sample of gamers by combining a relatively new source of data, actual behavioral footprint from game developers, with a traditional survey. The background information and justification is very well developed and the methods are strong. Such a difficult analysis is hard to describe with enough clarity that people who aren't familiar with the analysis can understand it, and some suggestions are made here for minor changes that could help that. I also have a slight concern about period effects and the description of the sample that are not listed in the Limitations section. In my opinion, the paper needs only minor revisions before it can be published and set a new standard in the field of video game research. Please see the attached file for many additional suggestions.

===PREPARING YOUR MANUSCRIPT===

===PREPARING YOUR REVISION IN SCHOLARONE===

Author's Response to Decision Letter for (RSOS-202049.R0)

See Appendix B.

RSOS-202049.R1 (Revision)

Review form: Reviewer 1 (Linda Kaye)

Is the manuscript scientifically sound in its present form?

Yes

Are the interpretations and conclusions justified by the results?

Yes

Is the language acceptable?

Yes

Do you have any ethical concerns with this paper?

No

Have you any concerns about statistical analyses in this paper?

No

Recommendation?

Accept as is

Comments to the Author(s)

The authors have done a good job of addressing the comments from reviewers. I am confident that this is to a publishable standard and look forward to seeing it in print

Review form: Reviewer 2 (Michelle Colder Carras)**Is the manuscript scientifically sound in its present form?**

Yes

Are the interpretations and conclusions justified by the results?

Yes

Is the language acceptable?

Yes

Do you have any ethical concerns with this paper?

No

Have you any concerns about statistical analyses in this paper?

No

Recommendation?

Accept as is

Comments to the Author(s)

Congratulations; I look forward to seeing this in print!

Decision letter (RSOS-202049.R1)

Dear Dr Johannes,

It is a pleasure to accept your manuscript entitled "Video game play is positively correlated with well-being" in its current form for publication in Royal Society Open Science. The comments of the reviewers who reviewed your manuscript are included at the foot of this letter.

Best regards,

on behalf of Professor Joydeep Bhattacharya (Associate Editor) and Essi Viding (Subject Editor)
openscience@royalsociety.org

Reviewer comments to Author:

Reviewer: 1
Comments to the Author(s)

The authors have done a good job of addressing the comments from reviewers. I am confident that this is to a publishable standard and look forward to seeing it in print

Reviewer: 2
Comments to the Author(s)

Congratulations; I look forward to seeing this in print!

Appendix A

Review of Video game play is positively correlated with well-being

This landmark paper examines well-being in a large sample of gamers by combining a novel source of data, actual behavioral footprint from game developer data, and a with traditional survey. The background information and justification is very well developed and the methods are strong. Such a difficult analysis is hard to describe with enough clarity that people who aren't familiar with the analysis can understand it, and some suggestions are made here for minor changes that could help that. I also have a slight concern about cohort effects and the description of the sample that are not listed in the Limitations section. In my opinion, the paper needs only minor revisions before it can be published and set a new standard in the field of video game research.

A few overall suggestions:

This is a very rigorous study. Using reporting criteria such as JARS-Quantitative or the STROBE checklist would improve transparency and standardize suggested elements of the paper. These could be included on the OSF website.

I think of conflict of interest as something that you declare if you have any kind of association with a conflicted partner, so I would recommend updating that section to be more transparent about the collaboration with industry partners. This is played up in the abstract and elsewhere, but then not described in the official conflict of interest statement. In other words, don't say there's no conflict of interest, just describe what could be viewed as a potential conflict. This might be stated in the conflict of interest section as "The study was conducted with data donated by video game developers. However, these companies had no say in study design, analysis, or reporting of results." (or whatever the case may be). This could also be clarified on page 2.

The word telemetry has a very specific meaning, especially as related to health. While it can be used to describe collection of data from software applications, it is more often reserved for remote monitoring and transfer of data from devices, e.g. between a patient's bedside and a nurses' station. I would recommend using a term used in similar research like captured behavioral data, behavioral markers (Liu paper), [device] use, digital footprint, data exhaust, etc.

Also, in a few places it is stated that previous research had to rely solely on self-reported play behavior. There is at least one well-known study from 2011 that was part of the Virtual Worlds Exploratorium project. In this study, researchers collaborated with Sony to capture data from Everquest II. I think a more in-depth exploration of this effort along with anything else by Williams, Yee, and/or the Virtual Worlds Exploratorium would better contextualize this work. There has also been more recent data from Pokemon GO and Solitaire, at the least. Some literature can also be found in the computer science field, e.g. ACM Digital Library.

Also please note that I have made extensive suggestions for improvement, mostly for clarification, but also in some cases for things that I think may represent bias that is not adequately discussed. Nonetheless, I think these will be straightforward to address.

Page 2

Unless the UNICEF 1990 citation talks about video games, I would use something out of Kardefelt Winther's UNICEF involvement, or the report by Livingstone instead. Here's a good UNICEF blog

post that describes this.

It would be good to contextualize the Digital, Culture, Media and Sport Committee. The only context I see is in the reference, which has .uk. Also, is the policy focus on time played or loot boxes? That should be clarified.

It is important to explain the link between subjective well-being and mental health if that claim is made. Several frameworks have been used to tie these together, e.g. flourishing.

p. 3

I think it's important to separate out the FDA-approved game from commercial off-the-shelf games designed for entertainment.

Scientists who have experience in combining such telemetry data-see Williams' company Ninjametrics

Data is not only self-reported, but also parent- and clinician-reported

These prior efforts to link captured data and self-report could be linked to IJzerman's evidence level

p. 4

When talking about the methodological challenges and needs for collaboration when trying to understand links between gaming and mental health, it may be useful to look at work by Colder Carras (therapeutic gaming), Scholten & Granic (design thinking), and Fleming (future directions for serious gaming)

I would describe the problem with video game effects as "clinically irrelevant", perhaps, and suggest we need measures of game time and other behavioral indicators of play.

p.5

I would say lower psychological functioning rather than decreased

for those who are new to Ryan, Przybylski, etc on extrinsic motivation, some additional explanation would be helpful. What does it mean to feel pressured to play?

p. 6

What are the implications for not incentivizing participation? Especially when the survey was open for such a short time . Note the number of responses in the Shen and Williams paper when the survey was open for much longer and the incentive was a unique in-game item

Please describe how the data ID was hashed

Here are some things to consider about the sampling method:

-How were the 50,000 players chosen from all possible players? How was the second wave chosen? Why was the response time for the survey so short?

-How did you confirm that the data was suitable on page 6?

-How was the response time for the second wave chosen?

-Why was the number of matching behavioral data so much smaller and what do you make of having so few women in the PvZ sample?

-Why were two different survey platforms chosen?

-Were the Nintendo invitations sent via email as well? Why was the response window only seven days for that one? Why do you think there were so many more women in the second group?

p 7

Under measures who was the Scale of Positive and Negative Experiences normed on? Is the way that it was treated in the study the way that is normally treated?

I really like the graphical displays of well-being

p. 8

I looked up the PENS and I wasn't able to find something that matched exactly, including the manner of play or the enjoyment and extrinsic motivation. Could these be explained and cited?

Were screen time reports available for those two games? If so, if you do similar studies, I wonder if you should have asked players to include that.

The description of overlapping game times is a pretty confusing. It might help to have some kind of diagram or graph.

Does aggregation work when you are essentially creating proxies for separate sessions? I'm not sure if that's covered adequately in the limitations-it's still a bit confusing. How did Nintendo verify the session duration? And why was the two week period chosen for aggregation?

p. 9

Could you please report a table of descriptive statistics for the variables in the analysis?

It would help if you included titles (e.g., actual game time per week) for the two parts of Fig 2 rather than just putting the description of the two graphs in the notes. Otherwise, lovely graph.

In the notes, please the % of truncated values and clarify if what the highest truncated value was or some other indication of the extremity of the truncated values.

Please describe the analyses conducted and how missing data was treated.

p. 10

Why are the N's described in the figures different from the total sample size? I imagine it's because cases that were missing data on the covariates were dropped from the regression. Is there any reason to think the missingness is MCAR?

How do you interpret the effect sizes of wellbeing effects? Have these effects sizes translated into

relevant impacts on wellbeing or mental health in previous studies? This could use a bit more to contextualize it, especially given that this section is full of numbers. Another option would be to add a table and just give the description of the results in the text (e.g., summarizing what's going on).

There's something strange going on in 3C with the high values of playtime that warrants comment.

p 11

Does the regression include all of those variables together, run separately by game? Are there any control variables (e.g., age or gender)? Or are there two separate models being described?

I think it's a bit misleading to include competence in the list of variables that predicted wellbeing when the Cis overlapped 0 for both games. On the other hand, I think Intrinsic could be described as significant, but only in 1 game.

To promote standardized reporting for later meta-analyses, please include measures of effect size and precision (see earlier recommendation about JARS-Quant or STROBE reporting checklists).

p 12

I think it would be more accurate to characterize your main results by what you did find in the regression model rather than what you didn't find: motivations to play affected wellbeing, regardless of hours.

Is the policy-relevant piece that games should be regulated by time? Who is proposing that (other than professional organizations like the AAP)? If there's a regulatory body proposing it, perhaps it would be relevant to bring in literature related to how ineffective shutdown laws are.

p13

It's hard to imagine that play time has a completely linear association with wellbeing, given that values were truncated at 80 hpw. I realize there was a statistical analysis to check for nonlinear relationships, but it seems to lack face validity. I wonder if this could be compared to other studies of wellbeing or problematic gaming and time playing.

One other important thought: Could this unusual linear relationship be a period effect due to the pandemic? Perhaps at the time the data were gathered, the relationship was linear because gamers were using games to cope. This should be considered.

The discussion of effect sizes and their relationship to perceived changes in wellbeing is lacking the link to how these small effect sizes might lead to changes in mental health.

I think it then would be appropriate to qualify statements like "our results suggest that play can be an activity that relates positively to people's mental health" by (1) discussing the pandemic/period effects and (2) clearly justifying the link between wellbeing and mental health.

p14

"The opposite was the case" – seems like it might mean "it was not positively related", not "it was negatively related".

The bit about player experiences as a moderating role should be moved up closer to the beginning of the discussion, as these seem to be primary findings.

Did the model have adequate power to detect effects with all those covariates?

Again, if the goal was to look at mental health, it would be good to clarify how the scale used correlates with measured mental health under some framework or theory.

In the limitations, there should be about a paragraph talking about the participants, recruitment, implications about the period in which data was collected, the sample, and missing data.

p15

Some game companies and platforms make their APIs accessible. These have been used in previous research outside of psychology and might be worth investigating/mentioning, as well as the older studies, e.g. the Virtual Worlds Exploratorium.

It's still really hard to understand the data processing problems. A table or diagram would help, as this is vital to understanding the implications of the methods and findings.

Appendix B

Revision for RSOS-202049 "Video game play is positively correlated with well-being"

Dear editorial team,

Thank you for giving us the opportunity to revise our manuscript. We are also grateful for the constructive and helpful feedback from the two reviewers. Their comments helped us make the manuscript stronger. We reply to each point they raise below. Working in their feedback has made the manuscript better and we are confident it now meets the quality standards for publication in *Royal Society: Open Science*. Please note that reviewer 2's comment (#35) on the sample sizes per analysis made us go back to the analysis where we found a small bug in the code. We corrected all numbers in the manuscript and the OSM. The bug did not influence the conclusions.

Reviewer 1

1. Thank you for providing me the opportunity to review this work. I really like the approach of using objective play time data to correspond to well-being measures. This is a much needed development and contribution in this field. The work has been conducted in a rigorous way and presented well. Thank you to the authors for providing their resources on OSF and the offer of using the supplementary data. I have some minor comments which may support some of the conceptual framing/presentation of the manuscript and overall am highly supportive of this work being published.

Thank you for your kind words. We are happy to hear that you find that the manuscript makes a contribution to the literature. Thank you also for your constructive and helpful feedback.

2. The introduction could do with **a little more theoretical detail about basic needs theory/SDT** as currently player needs is discussed only in passing. What are these and why are they relevant to gaming?

This is an excellent point. In the initial submission, we aimed to keep the introduction as brief as possible. However, your comment makes it clear that we should provide a more in-depth framing of self-determination theory so that readers who are not familiar with that theory might better understand our inclusion of measures based on this part of the human motivation literature. In line with this point we have re-drafted the paragraph on the role of motivation on gaming, and added a new, short paragraph on self-determination theory right before the initial paragraph. Here, we also address your fourth point on the rationale behind the moderation (p 5-6):

“Whereas the perceptions of players in recalling their video game play time can introduce bias, a decade of research indicates perceptions of the psychological affordances provided by games are important to player experiences in games. According to self-determination theory, any activity whose affordances align with the motivations of people will contribute to their well-being [50]. Motivations can be intrinsic, driven by people's interests and values which result in enjoyment, or extrinsic, inspired by rewards or a feeling of being pressured to do an activity. If an activity also satisfies basic psychological needs for competence, relatedness, and autonomy, people will find the activity more motivating, enjoyable, and immersive – ultimately leading to higher well-being.

The interplay of the affordances of video games, motivation, and needs has shown to be important for subjective well-being. If a game satisfies basic needs people will experience more enjoyment and higher

well-being [51]. Conversely, if those needs are not met, frustrated, or play is externally motivated, it is associated with lower psychological functioning [52]. In other words, how play time relates to well-being likely depends on players' motivations and how the game satisfies basic needs. Player experience would thus moderate the association between play time and well-being: If players are intrinsically motivated and experience enjoyment during play, play time will most likely be positively associated with well-being [53,54]. . . . “

3. The authors talk about “well-being” and then mention example studies, one including depression. **As well-being is a wide construct, some outline of this would be useful** ie how are the authors conceptualising it when discussing it within the introductory sections? Also, in the “This study” section, the authors interchange the terminology. In some cases they outline they will be exploring “subjective well-being” and other times “subjective mental health”. Are these the same thing? The method suggests that affective wellbeing was measured so some conceptualisation earlier in the paper would be helpful on this.

Thank you for pointing this out. Indeed, we were inconsistent in how we framed the concept of well-being in the originally submitted manuscript. We now revised the introduction to better focus on the conceptualization of well-being (p 5):

“Research and policymakers have been interested in a wide range of mental health outcomes of video game play. Mental health comprises both negative mental health (e.g., depression) and positive mental health. Positive mental health can be further divided into emotional well-being (i.e., the affective component) and evaluative well-being (i.e., the cognitive component) [38]. Nearly all non-experimental studies examining the links between video games and mental health rely on subjective, self-reported estimates of video game time, either by players themselves or by parents.”

We also made sure to stay consistent in our terminology.

4. The analysis discusses that the authors paid attention to the moderating role of needs and motivations between play time and wellbeing. There has been no conceptual framing of this previously and so the intro would benefit from some detail about this. Eg **in what way are specific needs likely to be a moderator between these variables?** There is a brief description of expecting findings on p11, but no theoretical/empirical basis for this in the intro in the current manuscript.

Thank you for pointing this out. Indeed, we introduced the rationale for the moderation tests quite late in the paper. Now, we introduce it earlier in the introduction, namely in the revised section on self-determination theory (see our response to your second point).

5. The current study focuses exclusively on two games as is appropriate given the methodology of garnering specific game server data. The intro prior to this has discussed gaming in very generic terms and I wonder how much of the evidence in the literature review is relevant to the types of games being used in the current work. My understanding from this literature is a lot of what has been discussed in the lit review relates more to studies which have looked at MMOs or assumed “hardcore” forms of gaming (or been very general). The two games in the current study are a simulation game and TPS, both of which seem more tailored to the casual market to some extent. **How applicable are the previous findings about gaming and well-being to the specific**

types of games of the current study? It would be useful for the authors to outline some distinctions here in their rationale-building perhaps to identify that these specific types of games may have been less exclusively focused on in the previous literature (and this itself may be a contribution). This may have implications however, as the links between gaming and wellbeing found here may not preclude the previous work on different types of games and well-being being invalid. The authors do make reference to the issue of game genre in the discussion limitations **but it would be useful that this is framed earlier in the paper**

We are in full agreement that the data we report on the two games we studied do not represent the wide breath of video game experiences which might be studied. Understanding that we could have introduced concepts such as video game genre early in the paper we decided not to because considerations such as these were not part of our rationale for conducting the study or identifying our research questions. That said, the reviewer is correct that structure and function of digital spaces is important to study and they do have implications for gauging the generalisability of our findings. In line with this we have expanded our discussion of limitations to be precise on this point.

“Selecting two titles out of a wide range of games puts further limitations on how generalizable our results are. Especially *Animal Crossing: New Horizons* is considered a casual game with little competition. Therefore, although those two titles were drawn from different genres, we cannot generalize to players across all types of games [73]. The results might be different for more competitive games. Different games have different affordances [74] and, therefore, likely different associations with well-being.”

6. P9-10- would use the term “play time” consistently rather than switch between “game time” and “play time” (see also discussion for this too)

We agree that being consistent with the terms is important, and now use the term “play time” consistently throughout the revised manuscript. Thank you for pointing this out to us.

Reviewer 2

1. This landmark paper examines well-being in a large sample of gamers by combining a relatively new source of data, actual behavioral footprint from game developers, with a traditional survey. The background information and justification is very well developed and the methods are strong. Such a difficult analysis is hard to describe with enough clarity that people who aren't familiar with the analysis can understand it, and some suggestions are made here for minor changes that could help that. I also have a slight concern about period effects and the description of the sample that are not listed in the Limitations section. In my opinion, the paper needs only minor revisions before it can be published and set a new standard in the field of video game research. Please see the attached file for many additional suggestions.

Thank you for complimenting our efforts and approach. We appreciate your detailed feedback. We followed most of your suggestions and believe it made the paper stronger.

2. This is a very rigorous study. **Using reporting criteria such as JARS-Quantitative or the STROBE checklist** would improve transparency and standardize suggested elements of the paper. These could be included on the OSF website.

When we wrote up the study, we aimed to be as detailed as possible and include all elements of the STROBE checklist. However, your comments made it clear to us that we needed more details, especially on missing values and the sample sizes per analysis. Please see our responses to those points. The manuscript now describes all elements of the STROBE checklist or explicitly refers readers to the appropriate section in the online supplementary materials where more details are needed. We think that including a separate checklist document would not be beneficial to communicating our research concisely and with clarity.

3. I think of conflict of interest as something that you declare if you have any kind of association with a conflicted partner, so **I would recommend updating that section to be more transparent about the collaboration with industry partners**. This is played up in the abstract and elsewhere, but then not described in the official conflict of interest statement. In other words, don't say there's no conflict of interest, just describe what could be viewed as a potential conflict. This might be stated in the conflict of interest section as "The study was conducted with data donated by video game developers. However, these companies had no say in study design, analysis, or reporting of results." (or whatever the case may be). This could also be clarified on page 2.

We agree with you that declaring any potential conflicts of interests is key to providing context for readers. This information was included in our original manuscript submission under author notes. It is possible this was abridged. It read "The funders had no role in study design, data collection and analysis, decision to publish, or preparation of the manuscript and the authors declare no conflicts of interest." Maybe that was not forwarded to the reviewer?

4. The **word telemetry has a very specific meaning**, especially as related to health. While it can be used to describe collection of data from software applications, it is more often reserved for remote monitoring and transfer of data from devices, e.g. between a patient's bedside and a nurses' station. I would recommend using a term used in similar research like captured behavioral data, behavioral markers (Liu paper), [device] use, digital footprint, data exhaust, etc.

We chose this term for several reasons. First, it is regularly used in data intensive industries, for example, online advertising, social media, and also games. Second, it was the term our industry partners used. To avoid any confusion, we repeated the clarification from the abstract "(i.e., logged game play)" at the first mention of telemetry on page 2.

5. Also, in a few places it is stated that previous research had to rely solely on self-reported play behavior. There is at least one well-known study from 2011 that was part of the Virtual Worlds Exploratorium project. In this study, researchers collaborated with Sony to capture data from Everquest II. I think **a more in-depth exploration of this effort** along with anything else by Williams, Yee, and/or the Virtual Worlds Exploratorium would better contextualize this work. There has also been more recent data from Pokemon GO and Solitaire, at the least. Some literature can also be found in the computer science field, e.g. ACM Digital Library.

Thank you for bringing this to our attention. Yes, we indeed were not clear. There have been previous studies that combined logged data with survey responses. These studies had different research questions. Notably, they did not investigate well-being and/or had accurate measures of game time. On page 5, we now cite and address these studies to provide better context for readers:

“A handful of pioneering studies has combined server logs with survey data [47]. However, these studies mainly took a network approach, modelling offline to online dynamics in leadership [48] and friendship formation in games [49]. Studies combining objective play and well-being are lacking. We need accurate, direct measures of play time to resolve the inconsistencies in the literature on well-being and to ensure the study of games and health is not as fruitless as the study of games and aggression [5].”

Page 2

6. Unless the UNICEF 1990 citation talks about video games, I would use something out of Kardefelt Winther's UNICEF involvement, or the report by Livingstone instead. Here's a good UNICEF blog post that describes this.

Thank you for the helpful suggestion. We were indeed trying to emphasize that restrictions might violate the human right to play and freedom of expression (i.e. UNCRC). On page 2, we now cite the UNICEF report and make that point clearer: “enacting policies that unnecessarily regulate play would restrict human rights to play and freedom of expression [3].”

7. It would be good to contextualize the Digital, Culture, Media and Sport Committee. The only context I see is in the reference, which has .uk. Also, is the policy focus on time played or loot boxes? That should be clarified.

We added “e.g. in the UK” to that reference. The report focused on addictive and immersive technologies, including online games, monetization models, and VR.

8. It is important to explain the link between subjective well-being and mental health if that claim is made. Several frameworks have been used to tie these together, e.g. flourishing.

This comment aligns with a comment of reviewer 1. On page 5 (see responses above), we briefly explain the concepts of mental health and well-being and cite a review paper by Diener et al. that also expands on eudaimonic parts of well-being, such as flourishing.

p. 3

9. I think it's important to separate out the FDA-approved game from commercial off-the-shelf games designed for entertainment.

We agree that this distinction is important. On page 3, we make clear that the FDA approved the use of “a so-called ‘serious video game’” to distinguish it from commercial games.

10. Scientists who have experience in combining such telemetry data-see Williams' company Ninjametrics

On page 5, we cite and briefly detail Williams’ work on the Virtual Worlds Exploratorium and MMOs (see our response to point 5).

11. Data is not only self-reported, but also parent- and clinician-reported

We agree that this clarification is necessary. We added a half sentence on page 5: “self-reported estimates of video game time, either by players themselves or by parents”.

12. These prior efforts to link captured data and self-report could be linked to IJzerman's evidence level

We thought about linking our discussion about the policy implications of psychological work on games to that work, but felt the existing literature is nowhere near the level of Evidence Readiness Level (ERL) 1 and therefore did not add to the manuscript and decided against invoking ERLs.

p. 4

13. When talking about the methodological challenges and needs for collaboration when trying to understand links between gaming and mental health, it may be useful to look at work by Colder Carras (therapeutic gaming), Scholten & Granic (design thinking), and Fleming (future directions for serious gaming)

We acknowledge the need of recording in-game behaviors when we discuss how companies need to have accessible APIs for researchers on page 20: "Such in-game behaviours also carry much promise for studying the therapeutic effects of games, for example, as markers of symptom strength in disorders [76]. In rare cases, researchers were able to make use of such APIs [47,49], but the majority of gaming data are still not accessible."

14. I would describe the problem with video game effects as "clinically irrelevant", perhaps, and suggest we need measures of game time and other behavioral indicators of play.

When we discuss effect sizes on page 17, we added the sentence: "From a clinical perspective, it is likely the effect is too small to be relevant for clinical treatments."

p.5

15. I would say lower psychological functioning rather than decreased

We adjusted the wording on page 6.

16. For those who are new to Ryan, Przybylski, etc on extrinsic motivation, some additional explanation would be helpful. What does it mean to feel pressured to play?

This mirrors a suggestion from reviewer 1. We revised the sections on self-determination theory on pages 5-6 (see our response to reviewer 1, point 5).

p. 6

17. What are the implications for not incentivizing participation? Especially when the survey was open for such a short time. Note the number of responses in the Shen and Williams paper when the survey was open for much longer and the incentive was a unique in-game item

We do not think that the interaction between not incentivizing participation and response window played a significant role in our results, because across the three surveys (two windows for PvZ, one for ACNH) the survey windows were quite different, but the response rates remained relatively stable. The vast majority of responses were given very shortly after the email advertisements to the studies. It is however possible that not incentivizing might lead to unrepresentative estimates, both in our study and in the field more generally, because then prospective participants' financial situation might lead to differential participation rates. However, the same can occur with incentivized surveys, where the

motives of the survey respondents could differ as a consequence. Nevertheless, we now include a comment on potential selection effects in the revised manuscript (p 19):

“Another limiting factor on the confidence in our results is the low response rate observed in both of our surveys. It is possible that various selection effects might have led to unrepresentative estimates of well-being, gaming quantity, or their relationship. Increasing response rates, while at the same time ensuring samples’ representativeness, remains a challenge for future studies in this field.”

18. Please describe how the data ID was hashed

We do not know the exact process of hashing the ID because the industry partners were responsible for that part. With EA, they sent out the emails on their own survey platform, hashed the player ID internally, and sent us the data. With Nintendo of America, we provided the link to the formr survey. Nintendo then sent this survey to the players, with a URL that captured a hashed player ID.

19. How were the 50,000 players chosen from all possible players? How was the second wave chosen? Why was the response time for the survey so short?

We decided to send out two waves to EA players for two reasons: First, we wanted to gauge the response rate and thereby determine the size of the second wave. Second, we wanted to ensure the data linking works before sending the survey to the full sample. We agreed on 50,000 players to ensure that we would get at least a couple of responses to try out matching telemetry with the survey (based on our best guess of the response rate). EA sent out the survey to players who had been at least somewhat active in the past weeks. The response window was relatively short to reduce the influence of contextual factors. However, we learned, on the basis of the low response rate, that we needed to extend the response window.

20. How did you confirm that the data was suitable on page 6?

We carried out basic checks, such as whether we could match telemetry with surveys, whether there were implausibly large values etc.

21. How was the response time for the second wave chosen?

We wanted to strike a balance between reducing contextual factors and response rate, so we doubled the response window of the first wave.

22. Why was the number of matching behavioral data so much smaller and what do you make of having so few women in the PvZ sample?

This can be attributed to the differences in measuring telemetry for the two companies. The survey was sent to active players, but that does not mean that each player also played in the two preceding weeks. As for gender, it is possible that there are more women playing Animal Crossing than Plants vs. Zombies, but that is merely our speculation.

23. Why were two different survey platforms chosen?

We chose different survey platforms for logistical reasons. One partner already had a survey workflow in place, whereas the other could work with the survey platform we chose.

24. Were the Nintendo invitations sent via email as well? Why was the response window only seven days for that one? Why do you think there were so many more women in the second group?

We learned from the response window in the collaboration with EA. Again, we wanted to strike a balance between reducing the influence of contextual factors while getting a larger response rate. As for gender, see our speculation on point 22.

p 7

25. Under measures who was the Scale of Positive and Negative Experiences normed on? Is the way that it was treated in the study the way that is normally treated?

The scale has been validated, but not normed. Usually, the scale is used as a sum score. To accommodate missing values, we took the mean of the scale, which is not biased by missing values unlike the sum score.

26. I really like the graphical displays of well-being

Thank you, we thought readers would prefer a graphical display rather than a table of numbers.

p. 8

27. I looked up the PENS and I wasn't able to find something that matched exactly, including the manner of play or the enjoyment and extrinsic motivation. Could these be explained and cited?

The PENS scale is proprietary (we cite the developers of the scale). You can inspect the items on the OSF, where we provide all items we used.

28. Were screen time reports available for those two games? If so, if you do similar studies, I wonder if you should have asked players to include that.

The game time we report is as close to a meaningful screen time report as possible. We included all sessions, which included time spent in the hub world for Plants vs. Zombies.

29. The description of overlapping game times is a pretty confusing. It might help to have some kind of diagram or graph.

We wanted to keep the numbers of figures as low as possible. On the Github page, there are several graphs that show the overlap in sessions and computer code detailing the issue and how we resolved it. If the editor also believes they are necessary, we are happy to include them in the paper.

30. Does aggregation work when you are essentially creating proxies for separate sessions? I'm not sure if that's covered adequately in the limitations-it's still a bit confusing. How did Nintendo verify the session duration? And why was the two week period chosen for aggregation?

I think our description might have been confusing here. We did not create proxies for sessions, but aggregated overlapping ones: If two sessions had the same start time, but different end times, we took the start time and the latest end time. Afterwards, we summed up the durations of all game sessions within the two week window preceding the survey. Unfortunately, we cannot speak to Nintendo's internal logging procedures. We chose the two-week window to a) ensure the same time frame for both

studies, b) make sure we would capture enough play. The two week window corresponded to the survey questions, that asked about play and motivations in the preceding two weeks.

p. 9

31. Could you please report a table of descriptive statistics for the variables in the analysis?

We wanted to avoid overwhelming readers with too many numbers, which is why we chose figures, which are more informative than tables. However, descriptive statistics are available in the online supplementary materials: <https://digital-wellbeing.github.io/gametime/describe-explore.html#create-summary-figure>, and additional ones can be computed from the raw data that is provided.

32. It would help if you included titles (e.g., actual game time per week) for the two parts of Fig 2 rather than just putting the description of the two graphs in the notes. Otherwise, lovely graph.

We have now indicated the game title and variable type (actual time vs estimated time) in the legend of Figure 2.

33. In the notes, please the % of truncated values and clarify if what the highest truncated value was or some other indication of the extremity of the truncated values.

Thank you for the suggestion. We included those details in the figure caption.

34. Please describe the analyses conducted and how missing data was treated.

In the statistical analysis section, we explain that the sample size per analysis will differ because of missing data, exclusions, or lack of social play (see point below).

p. 10

35. Why are the N's described in the figures different from the total sample size? I imagine it's because cases that were missing data on the covariates were dropped from the regression. Is there any reason to think the missingness is MCAR?

Thank you for that suggestion. Indeed, the sample sizes differ depending on missing values. We added an explanation under statistical analysis: "Note that the sample sizes per analysis will differ because of missing values either due to exclusions, not having telemetry data, or not having played socially (i.e., no responses on relatedness)."

In response to this comment, we went back to the code and found a small bug that resulted in too many missing values on the estimated game time: When participants reported only minutes or only hours, the sum in the previous code version resulted in a missing value. The correction resulted in larger samples for analyses on subjective game time and slightly changed numerical estimates. Substantive conclusions remained the same, but because there is now less missing data, the resulting estimates are more informative and thus led to an overall stronger manuscript.

As for the missingness pattern, it is almost certain that missingness on telemetry is MCAR. As for the missing responses on relatedness items, this might yet again be subject to a selection effect, see new section in limitations.

36. How do you interpret the effect sizes of wellbeing effects? Have these effects sizes translated into relevant impacts on wellbeing or mental health in previous studies? This could use a bit more to contextualize it, especially given that this section is full of numbers. Another option would be to add a table and just give the description of the results in the text (e.g., summarizing what's going on).

We agree that the section is full of numbers, which is why we tried to visualize all parameters. As such, we provided tables of all coefficients on the Github page, but did not want to include it in the manuscript, because tables are less effective in communicating information than are graphical depictions. We refrained from discussing effect sizes in the results section, but dedicated an entire section to the smallest effect size of interest in the discussion, where we also place the estimates in context.

37. There's something strange going on in 3C with the high values of playtime that warrants comment.

Indeed, self-reported hours from EA had a cut-off of 40h 59min, which is likely due to the survey formatting. However, extremely few cases got close to that cut-off. We added more details to the manuscript on page 10: "For PvZ, the *Decipher* survey platform restricted the maximum time players could report to 40h 59 min. Such large values were rare and affected only a handful of participants. The AC:NH time scale was unrestricted."

p 11

38. Does the regression include all of those variables together, run separately by game? Are there any control variables (e.g., age or gender)? Or are there two separate models being described?

Yes, the models are exactly as the reviewer describes: All variables are entered together, with one model per game. We did not include covariates, simply because we are not aware of a directed acyclic graph that would warrant those controls.

39. I think it's a bit misleading to include competence in the list of variables that predicted wellbeing when the Cis overlapped 0 for both games. On the other hand, I think Intrinsic could be described as significant, but only in 1 game.

That sentence only describes the direction of the association. In the following sentence we make clear that this association was not significant for competence. If the editor thinks that listing competence here is misleading, we are happy to only focus on significant predictors.

40. To promote standardized reporting for later meta-analyses, please include measures of effect size and precision (see earlier recommendation about JARS-Quant or STROBE reporting checklists).

We agree that effect size and precision estimates are important, and have therefore included them in the manuscript. In the text of the results, we report standardized effects (regression betas) because the outcome has been standardized. There, we also report confidence intervals for those estimates, from which the standard errors of the estimates can be computed. The full numerical output is on Github: <https://digital-wellbeing.github.io/gametime/analyses.html#objective-time-and-swb> To make clear to readers that they can find the full regression table online, we added a sentence on page 12: "Readers

can find the full regression tables on <https://digital-wellbeing.github.io/gametime/analyses.html#analyses>.”

p 12

41. I think it would be more accurate to characterize your main results by what you did find in the regression model rather than what you didn't find: motivations to play affected wellbeing, regardless of hours.

We added the following sentence to the discussion on page 16: “We did not find evidence that this relation was moderated by need satisfactions and motivations, but that need satisfaction and motivations were related to well-being in their own right.”

42. Is the policy-relevant piece that games should be regulated by time? Who is proposing that (other than professional organizations like the AAP)? If there's a regulatory body proposing it, perhaps it would be relevant to bring in literature related to how ineffective shutdown laws are.

The Digital, Culture, Media and Sports Committee considered the question of regulating game time, but we didn't think expanding greatly on this topic would benefit the manuscript. We do not believe that the literature, or indeed our study, is well positioned to provide positive policy or regulatory advice.

p13

43. It's hard to imagine that play time has a completely linear association with wellbeing, given that values were truncated at 80 hpw. I realize there was a statistical analysis to check for nonlinear relationships, but it seems to lack face validity. I wonder if this could be compared to other studies of wellbeing or problematic gaming and time playing.

We were not clear here: Values were only truncated for visualization (the revised figure caption now says that the values were truncated “... in this figure”, but not for the analysis. For the analysis, we used the full data set. We are not sure how the generalized additive model we used to check for nonlinear associations lacks face validity because it can take any form of relationship indicated by the data.

44. One other important thought: Could this unusual linear relationship be a period effect due to the pandemic? Perhaps at the time the data were gathered, the relationship was linear because gamers were using games to cope. This should be considered.

We agree that the pandemic could certainly influence the results. We now discuss the possibility of periodic contextual factors in an updated paragraph in the limitations section on page 19:

“Longitudinal work would also address the question of how generalizable our findings are. We collected data during a pandemic. It is possible the positive association between game time and well-being we observed only holds during a time when people are naturally playing more and have less opportunity to follow other hobbies. However, recent work on the relation between technology use and well-being over time shows that the relation is remarkably stable.”

45. The discussion of effect sizes and their relationship to perceived changes in wellbeing is lacking the link to how these small effect sizes might lead to changes in mental health.

This mirrors a concern from reviewer 1 on defining the concept of mental health. On page 5, we now define mental health and what facet we investigate. In the discussion on effect sizes, we then added a clarification that effect sizes cannot be transferred to other facets: "Moreover, we only studied one facet of positive mental health, namely affective well-being. Future research will need to consider other facets, such as negative mental health."

46. I think it then would be appropriate to qualify statements like "our results suggest that play can be an activity that relates positively to people's mental health" by (1) discussing the pandemic/period effects and (2) clearly justifying the link between wellbeing and mental health.

The statement the reviewer refers to summarizes the project. We believe the updated limitations section (see responses above) addresses and qualifies that summary.

p14

47. "The opposite was the case" – seems like it might mean "it was not positively related", not "it was negatively related".

We agree that the sentence is not clear. We updated it to: "Extrinsic motivation, by contrast, was negatively associated with well-being."

48. The bit about player experiences as a moderating role should be moved up closer to the beginning of the discussion, as these seem to be primary findings.

Our main goal for this project was to study the relation between game time and well-being. Therefore, we introduced it first and also make it the first point of discussion. Even though the player experiences turned out to be stronger predictors of well-being, we wanted to avoid writing the paper following the results, and not the primary research question.

49. Did the model have adequate power to detect effects with all those covariates?

That is an open question and naturally depends on the assumed magnitude of the true relationship. Our sample was sensitive enough to detect a correlation of .06 with 80% power. Interactions require greater samples for such an effect size, but we would argue that the coefficients were this close to zero that they were clearly below a smallest effect size of interest.

50. Again, if the goal was to look at mental health, it would be good to clarify how the scale used correlates with measured mental health under some framework or theory.

This echoes the concern of reviewer 1. We defined and clarified mental health on page 5.

51. In the limitations, there should be about a paragraph talking about the participants, recruitment, implications about the period in which data was collected, the sample, and missing data.

Thank you for the great suggestion. We believe the new paragraph on possible selection effects addresses this concern (page 19).

p15

52. Some game companies and platforms make their APIs accessible. These have been used in previous research outside of psychology and might be worth investigating/mentioning, as well as the older studies, e.g. the Virtual Worlds Exploratorium.

In the discussion, we now explain that there have been a few studies that have used APIs, but that such APIs are still too rare: “In rare cases, researchers were able to make use of such APIs [47,49], but the majority of gaming data are still not accessible.”

53. It's still really hard to understand the data processing problems. A table or diagram would help, as this is vital to understanding the implications of the methods and findings

We give a detailed overview and several graphs on the Github page. We now tell the reader that they can find that information there in the Method section: “Readers can find more details on data processing on <https://digital-wellbeing.github.io/gametime/>.”

Response Summary

We want to take this opportunity again to thank the editor and the reviewers for their extremely helpful feedback. We addressed all of their comments directly and through substantial revisions to the manuscript on three major elements: First, we expanded on self-determination theory (SDT) and provided more theoretical context for those readers who are not familiar with the framework. This included highlighting aspects of SDT that directly related to our investigation. Second, we provided more details on missing values and sample size per analysis to make it easier for readers to follow our procedure and to conform with the STROBE checklist. Third, we increased conceptual clarity for the construct of well-being and expanded the scope of the potential limitations to our conclusions.

We believe we were able to fully address the points that were raised. It is our hope that you will find the work fully and adequately revised and that you judge it suitable for publication in *Royal Society Open Science*.

With kind regards,

Niklas Johannes, Matti Vuorre, and Andrew K. Przybylski